# Design for SAW Antenna-Plexers with Improved Matching Inductance Circuits

**DOI:** 10.3390/mi15010089

**Published:** 2023-12-30

**Authors:** Min-Yuan Yang, Ruey-Beei Wu

**Affiliations:** Department of Electrical Engineering and Graduate Institute of Communication Engineering, National Taiwan University, Taipei 10617, Taiwan

**Keywords:** antenna-plexer, optimization, SAW extractor, diplexer, WIFI 6E

## Abstract

This study designs antenna-plexers, including a surface acoustic wave (SAW) extractor and an upper- and mid-high band (UHB + MHB) diplexer, for LTE 4G and 5G bands using carrier aggregation. The SAW extractor combines a bandpass filter (BPF) and a band-stop filter (BSF) in a single unit that consists of eight modified Butterworth–van Dyke (mBVD) resonators that resonate in parallel with an inductor and SAW resonators. This BSF behaves as a high-pass filter at frequencies lower than the designed WIFI band and as a capacitor at higher frequencies. The SAW extractor meets product specifications in the frequency range 0.7 to 2.7 GHz. The UHB + MHB diplexer, which is composed of a microwave filter, a SAW filter, and a simple matching inductor, uses frequency response methods to create an RF component for 2.4 GHz + WIFI 6E applications. The design uses a SAW’s interdigital transducer (IDT) structure, and the experimental results are in agreement with the simulation results, so the design is feasible.

## 1. Introduction

As the capabilities of modern cellular phones continue to increase, more bands must coexist with adjacent cellular frequencies, such as WIFI (2.4 GHz) and L5, between low and mid-high bands. Figure 1 shows the frequency bands that are used for handheld devices. These include satellite-based navigation systems (L5, L1) and low (700–900 MHz), mid (n3–n40), and high (n7, n41, n42, n43) bands for cellular communications as well as WIFI (2.4 and 5 GHz) [1].

A typical 4G smartphone has at least four to eight antennas, and 5G smartphones have more because they must support 5G bands and other bands, such as UWB/UHB, while still complying with all frequencies and standards set for 4G phones. As manufacturers integrate new features, such as cameras, facial recognition, and motion sensors, into 5G phones, the space that is available for antennas continues to shrink, so 5G handset manufacturers must make better use of the limited space that is allocated to RF components. 

Many studies propose methods to reduce the size of RF components. Acoustic wave (AW) filter modules are key components [2], but their performance degrades as frequency increases [3]. Compact RF passive components are realized by using low-temperature co-fired ceramic (LTCC) technologies, including diplexers [4,5,6,7,8,9,10], duplexers [11,12,13], band stop filters [14,15], and low pass filters [16]. However, when overall space constraints are considered, this may not be a suitable solution for a mobile phone.

Small-sized, high-efficiency RF modules in handsets with low impedance must be used to reduce the number of antennas that are required and to ensure that performance is maintained. Various applications use different frequency ranges, leading to diverse system requirements for each band. Crafting an effective filter module involves ensuring that its frequency response meets the specified requirements across all ranges. Therefore, intelligent design considerations, including topology selection, optimizing SAW resonator characteristic impedance, and careful material choice, are crucial [2]. This study uses an optimization approach [17] to design the antenna-plexer module, which is composed of surface acoustic wave (SAW) resonators and inductors. 

The remainder of the paper is organized as follows: Section 2 discusses the design process, including circuit topology, design objectives, and cost functions and constraints. The simulations and realization are demonstrated in Section 3 and Section 4, respectively, followed by the conclusions in Section 5.

## 2. Design Procedure

### 2.1. Circuit Topology

The circuit topology for a SAW extractor is shown in Figure 2. The SAW resonator is represented by the mBVD model [18] in Figure 3. WIFI port 2 is synthesized using a ladder-type structure that is shown in the upper channel in Figure 2. Port 3 connects to the cellular part. The matching circuit uses two inductors: a series inductor that is connected to antenna port 1 and a shunt inductor that is connected to port 3 in the lower channel of Figure 2.

The circuit topology for an upper- and mid-high band (UHB + MHB) diplexer is shown in Figure 4. Port 2 is connected to 2.4 GHz WIFI using a 7th-order ladder-type filter structure. Port 3 connects to the WIFI 6E filter. This architecture requires only a single inductor to perform matching.

### 2.2. Design Goals

In practical applications of SAW extractors, the cell frequency band has an attenuation level of less than −38 dB, and the passband portion needs to be greater than −1.5 dB in the 2.4 GHz WIFI band for the insertion loss ${Ŝ}_{21}$. Within the cellular band, the passband portion of the insertion loss ${Ŝ}_{31}$ must also exceed −1.5 dB, and the stopband portion must be less than −20 dB in the 2.4 GHz WIFI band. The return loss ${Ŝ}_{11}$ is typically less than −10 dB, as shown in Table 1. The WIFI band is abbreviated as in-band, with a frequency $f_{in}\;$ = 2.401–2.483 GHz, and the cellular band is abbreviated as out-band, with a frequency range of $f_{out}$ = 0.7–2.37 and 2.555–2.7 GHz [19].

For the UHB + MHB diplexer, in the WIFI band, ${Ŝ}_{11}$ must be less than −10 dB, the passband portion of ${Ŝ}_{21}$ must be greater than −1 dB, and the attenuation level of the remaining frequency bands must be lower than −35 dB, as shown in Table 2. The WIFI band is abbreviated as in-band, with a frequency $f_{in1}$ = 2.401–2.483 and 5.925–7.125 GHz, and the other bands are abbreviated as out-band, with a frequency range of **f**_out1_ = 1–2.4, 2.484–5.924, and 7.126–10 GHz [20].

### 2.3. Theoretical Analysis

To ensure that the frequency response meets the defined goals, a theoretical analysis is undertaken. In terms of circuit analysis, the *Z* matrix for the topology in Figure 2 and Figure 4 is solved using the mBVD model for each constituent SAW resonator.

According to the microwave circuits analysis [21], the impedance matrix [**Z**] for a multi-port network can be defined as
(1)$$\left[\begin{array}{c} V_{1}\\ V_{\begin{array}{c} 2\\ \vdots \end{array}}\\ V_{P} \end{array}\right]=\left[\begin{array}{ccc} Z_{11} & \cdots & Z_{1P}\\ \vdots & \ddots & \vdots \\ Z_{P1} & \cdots & Z_{PP} \end{array}\right]\left[\begin{array}{c} I_{1}\\ I_{\begin{array}{c} 2\\ \vdots \end{array}}\\ I_{P} \end{array}\right]$$
where $V_{i}$ and $I_{i}\;\left(i=1,\ldots ,\;P\right)$ are the voltage and current at the *i*-th port, respectively, and *P* is the number of ports. Each element in the impedance matrix can be observed from (1), and $Z_{ij}$ can be determined using the following expression:(2)$$Z_{ij}=\frac{V_{i}}{I_{j}}{｜}_{I_{k}=0\;and\;k\neq j}$$

By definition, it can be calculated by applying an input current $I_{j}$ to the *j*-th port while keeping all remaining ports open-circuited and measuring the open-circuit voltage at the *i*-th port.

The scattering parameter [**S**] matrix is defined in terms of the relationship between the incident voltage and the reflected voltage waves as
(3)$$\left[\begin{array}{c} {V_{1}}^{-}\\ {V_{\begin{array}{c} 2\\ \vdots \end{array}}}^{-}\\ {V_{P}}^{-} \end{array}\right]=\left[\begin{array}{ccc} S_{11} & \cdots & S_{1P}\\ \vdots & \ddots & \vdots \\ S_{P1} & \cdots & S_{PP} \end{array}\right]\left[\begin{array}{c} {V_{1}}^{+}\\ {V_{\begin{array}{c} 2\\ \vdots \end{array}}}^{+}\\ {V_{P}}^{+} \end{array}\right]$$
where $V_{i}^{+}$ and $V_{i}^{-}$ represent the voltage wave amplitudes that are respectively incident on and reflected from the *i*-th port. 

If the reference characteristic impedance $Z_{0i}$ for all ports is the same, *i.e.*, $Z_{0i}$ = $Z_{0}$, [**S**] is defined in terms of [**Z**] as [21]
(4)$$[S]={\left([Z]+Z_{0}[U]\right)}^{-1}\left(\left[Z\right]-Z_{0}\left[U\right]\right)\;$$
where $\left[U\right]$ is a unit matrix with diagonal elements that equal 1 and all others are zero.

### 2.4. Cost Functions and Constraints

To evaluate the performance of the antenna-plexer, the cost function is specified. Manufacturing limitations mean that the electromechanical coefficient of the substrate $k_{t}^{2}$ must be the same for all of the SAW resonators.

For practical specifications, the cost function is constructed in two parts: the in-band and out-band frequencies. Both entail subtracting the S-parameters over the frequency range of the antenna-plexers from the target specifications. If the result exceeds zero, the specification is not met, necessitating the cost function calculation. Use the pattern search method in MATLAB to iterate the calculation [18] until the result is satisfactorily close to zero, meaning it meets the specifications. The root-mean-squares-error (RMSE) is used for calculation. $N_{in}$, $N_{out}$, $N_{in1}$, and $N_{out1}\;$ are the number of frequency points for $f_{in}$, $f_{out}$, $f_{in1},$ and $f_{out1}$, respectively, at intervals of 1 MHz. The number of frequency point counts differ, so each part is divided by its respective point count. ${Ŝ}_{n1}$, $S_{n1\_in}$, $S_{n1\_out}$, $S_{n1\_in1}$, and $S_{n1\_out1}$ are measured in dB so that the cost function is defined as:(5)$$\mathrm{Cos}t=\sqrt{{Cost}_{S_{11}}}+\sqrt{{Cost}_{S_{21}}}+\ldots$$

For the SAW extractor, the cost function is:(6)$${Cost}_{S_{21}}\left\{\begin{array}{c} in=\frac{1}{N_{in}}\sum_{f\in f_{in}}{\left[max\left(0,\;{Ŝ}_{21}-S_{21\_in}\right)\right]}^{2}\\ out=\frac{1}{N_{out}}\sum_{f\in f_{out}}{\left[max\left(0,\;S_{21\_out}-{Ŝ}_{21}\right)\right]}^{2} \end{array}\right.\;$$
(7)$${Cost}_{S_{31}}\left\{\begin{array}{c} in=\frac{1}{N_{in}}\sum_{f\in f_{in}}{\left[max\left(0,\;S_{31\_in}-{Ŝ}_{31}\right)\right]}^{2}\\ out=\frac{1}{N_{out}}\sum_{f\in f_{out}}{\left[max\left(0,\;{Ŝ}_{31}-S_{31\_out}\right)\right]}^{2} \end{array}\right.\;$$
(8)$${Cost}_{S_{11}}\left\{\begin{array}{c} in=\frac{1}{N_{in}}\sum_{f\in f_{in}}{\left[max\left(0,\;S_{11\_in}-{Ŝ}_{11}\right)\right]}^{2}\\ out=\frac{1}{N_{out}}\sum_{f\in f_{out}}{\left[max\left(0,\;S_{11\_out}-{Ŝ}_{11}\right)\right]}^{2} \end{array}\right.\;$$

For the UHB + MHB diplexer, the cost function is:(9)$${Cost}_{S_{21}}\left\{out=\right.\;\frac{1}{N_{out1}}\sum_{f\in f_{out1}}{\left[max\left(0,\;S_{21\_out1}-{Ŝ}_{21}\right)\right]}^{2}$$
(10)$${Cost}_{S_{11}}\left\{in=\right.\;\frac{1}{N_{in1}}\sum_{f\in f_{in1}}{\left[max\left(0,\;S_{11\_in1}-{Ŝ}_{11}\right)\right]}^{2}\;$$

To ensure that the SAW resonators have a constant value for $k_{t}^{2}$, a constraint is enforced during the optimization process [17]
(11)$$k_{t}^{2}=1-{\left(\frac{f_{s}}{f_{p}}\right)}^{2}$$
where $f_{s}$ and $f_{p}$ are the series and parallel resonance frequencies, respectively, of the mBVD model in Figure 3.

### 2.5. Optimization Flowchart

Figure 5 shows a flowchart of the optimization algorithm for the antenna-plexer design, which considers the cost function within the constraints that are discussed in the preceding subsection. This study determines the BVD parameters that minimize the cost function. The specifications for the antenna-plexer are defined in terms of the system requirements to produce the results in Table 1 and Table 2.

The set of variables x that is used to optimize the matching circuit and each resonator is:(12)$$\mathrm{for}\;\mathrm{the}\;\mathrm{extractor}:\;x=x(L_{1},L_{2},Z_{m1\sim 8},\;f_{s1\sim 8})$$
(13)$$\mathrm{for}\;\mathrm{the}\;\mathrm{UHB}+\mathrm{MHB}\;\mathrm{diplexer}:\;x=x(L_{1},Z_{m1\sim 7},\;f_{s1\sim 7}$$

The series input impedance is defined as:(14)$$Z_{m}=\sqrt{L_{m}/C_{m}}\;$$
and the resonant frequency $f_{s}$ and the anti-resonant frequency $f_{p}$ correspond to the AW series and shunt resonators, respectively. Designing $Z_{m}$ and the resonant frequency requires adherence to two rules: the resonant frequency $f_{s}=$1/$\left(2\pi \sqrt{L_{m}\cdot C_{m}}\right)$ and $Z_{m}$ resides within the specified range of 50 to 10,000 Ohms. Given these two values, the inductance (*L*_*m*_) and capacitance (*C*_*m*_) can be obtained. In numerical optimization, $Z_{m}$= 50 Ohms is usually set as the initial value for iteration.

The selection of the initial guess for the extractor was performed as follows:(15)$$L_{1}=L_{2}=1\;nH,\;Z_{m1\sim 8}=Z_{0},\;f_{s1,s3,s5,s7}=f_{center},\;f_{s2,s4,s6,s8}=f_{center}\cdot \sqrt{1-k_{t}^{2}}.$$

The selection of the initial guess for the UHB + MHB diplexer was performed as follows:(16)$$L_{1}=1\;\mathrm{nH},\;Z_{m1\sim 7}=Z_{0},\;f_{s1,s3,s5,s7}=f_{center},\;f_{s2,s4,s6}=f_{center}\cdot \sqrt{1-k_{t}^{2}}$$
where
(17)$$Z_{0}=50\;\Omega \;,\;f_{center}=2.442\;GHz.$$

The desired mBVD parameters that meet the frequency response solution can be solved using a systematic optimization algorithm [22]. These are converted to various physical parameters for the SAW resonators [18]. The circuit topology is imported into Advanced Design System (ADS) to simulate the frequency response.

## 3. Simulation

The simulation results for the antenna-plexer are presented, and the BVD parameters for this design are shown in this section.

### 3.1. SAW Extractor

The eight SAW resonators and the matching circuits for the extractor in Figure 2 are optimized using the pattern search method [22]. Table 3 shows the mBVD model parameter values for the optimized extractor with two matching inductors: *L*_1_ = 4.7 nH and *L*_2_ = 6.8 nH.

Figure 6a shows the frequency response for $S_{21}$ in the optimized extractor. In the 2.4 GHz WIFI band, the
insertion loss to WIFI port 2 is less than 2 dB and to cellular port 3 is greater than 20 dB. Figure 6b shows the frequency responses for $S_{11}$ and $S_{23}$. The return loss in the antenna port in the 2.4 GHz WIFI band is greater than 10 dB, and the isolation between the two output ports is greater than 20 dB. The designed SAW extractor meets the product specifications that are described in Section 2.2.

### 3.2. UHB + MHB Diplexer

The seven SAW resonators and the matching circuits for the UHB + MHB diplexer in Figure 4 are optimized using the pattern search method [22]. Table 4 lists the mBVD model parameter values for the optimized diplexer with a matching inductor: *L* = 3.3 nH. 

Figure 7a shows the frequency response for $S_{21}$ in the optimized antenna-plexer. In the 2.4 GHz WIFI band, the insertion loss to WIFI port 2 is less than 3 dB and to WIFI 6E port 3 is less than 2 dB. Figure 7b shows the frequency responses for $S_{11}$ and $S_{23}$. There is only one inductor, so the return loss in the antenna port in the 2.4 GHz WIFI band ($S_{11}\;$< −8 dB) and WIFI 6E ($S_{11}\;$< −9 dB) is not ideal.

### 3.3. The Effect of the Inductor on Antenna-Plexer

The inductive effect in the matching circuits is studied. For the SAW extractor in Figure 2, the structure of the band-stop filter includes a series matching inductor and another parallel inductor between resonators 7 and 8. The optimization values are shown in Table 3. The design concept for this structure is described in the following paragraphs. 

If there are no series and parallel inductors, the SAW resonators act as a capacitor at high and low frequencies. If $k_{t}^{2}$ = 6.1%, the capacitance of the resonator is approximately 0.62 pF, so adding a parallel inductor creates a resonance with the capacitance and produces an open circuit at a frequency that is close to the design frequency of 2.442 GHz. This setup has the characteristics of a band-stop filter. 

Outside of the stop band, consider the frequency behavior of the filter in the low-frequency range of 0.7 to 2.3 GHz and the high-frequency range of 2.484 to 2.7 GHz. In both frequency ranges, resonators 6, 7, and 8 behave like capacitors. In the low-frequency range, resonator 7 in shunt with inductor *L*_2_ is inductive, and the setup behaves like a high-pass filter. In the high-frequency range, it behaves like a capacitor, and the setup behaves like a capacitor. Figure 8 shows a frequency response in which the performance is not ideal.

The Smith chart in Figure 9a shows that the capacitive effect is too prominent at close to 1.8 GHz. This is addressed by adding a series inductor to achieve an impedance of approximately 50 Ω at around 1.8 GHz. The Smith chart in Figure 9b shows that there is a significant improvement in matching in the higher-frequency range than at the design frequency. The overall frequency response is shown in Figure 10.

For the UHB + MHB diplexer, the matching circuit structure for this study is shown in Figure 4. It uses a parallel inductor near the antenna port. The value after optimization is shown in Table 4. The design concept for this matching circuit depends on the capacitive characteristics of SAW filters at high and low frequencies. SAW filters create a short circuit at high frequencies and an open circuit at low frequencies, as shown in Figure 11a. The WIFI 6E filter for this study also exhibits similar capacitive characteristics at low frequencies, as shown in Figure 11b.

However, there is a significant difference in capacitance between the WIFI 6E band and the 2.4 GHz WIFI band filters, so this architecture creates less than ideal matching conditions.

## 4. Experiment

### 4.1. Extraction for IDT Characteristics

Figure 12 shows the IDT structure of a single-port resonator. To extract the physical parameters of the structure, the experimental data and numerical curve fitting are required to predict the characteristics of a SAW resonator [22]. The relationship between pitch (*p*) and $f_{s}$ is plotted using the experimental data, which were measured by the manufacturer, and the curve is shown in Figure 13. The piezoelectric material for this study is ${42{}^{\circ}\;YXLiTaO}_{3}$, and the electrode thickness is 0.17 µm, the IDT pair is 200, the grating order is 20, the aperture length *W* is 50 µm, and the experimental data for the pitch width *p* of the resonator range from 0.89 µm to 0.74 µm (with an interval of 0.01 µm).

For the experimental data, the curve fitting is used to fit a first polynomial function. This is fitted as:(18)$$1.9521=f_{s}\cdot \;p$$
where the unit for $f_{s}$ is GHz and the unit for *p* is µm. The characteristics of a single-port SAW resonator are determined. This constraint is considered in the subsequent design stage.

### 4.2. Experiment Layout

To realize the SAW resonators for the antenna-plexer, the physical parameters for the interdigital transducer (IDT) of the SAW resonators in Figure 12 is determined. Given the number of IDT pairs ($N_{t}$) and gratings ($N_{p}$), the two main physical parameters that affect the structural characteristics are the pitch width (*p*) and the aperture (*W*). Using the method of a previous study [22] and Section IV.A., if $N_{t}$ = 200 and $N_{p}$ = 20, the values of *p* and *W* are listed in Table 5. Similarly, the values for the physical parameters of the IDT structure for the diplexer are determined and listed in Table 6.

Given the parameter values for the IDT structure in Table 5 and Table 6, the layout circuits are configured using the topologies in Figure 2 and Figure 4. Figure 14a,b show the top view layout, with a circuit area of 1.375 mm × 0.985 mm. The device-under-test (DUT) is then packaged to become the prototype module. The packaged modules for the SAW extractor and the UHB + MHB diplexer are respectively shown in Figure 15a,b.

### 4.3. Experiment Results

In the experiment, the SOLT calibration method is used to calibrate the vector network analyzer (VNA). The VNA is then connected with a cable to the packaged experimental prototype. Therefore, the measurement results include the degradations and other parasitic effects from interconnects in the package and transmission lines in the test board. For the SAW extractor, the product specifications are shown in Figure 16 and Table 7, and the $S_{31}$ performance for the stopband (WIFI Band) is excellent. 

However, there is a 2 dB difference between the simulated results and the measurement for the passband (cell band). For $S_{21}$, the simulated results and the measurement for the passband (WIFI band) differ by 1.5 dB, and the stopband (cell band) shows good performance ($S_{21}$ is −3.5 dB in the passband and −37 dB in the stopband. $S_{31}$ is −21 dB in the stopband and −2 dB in the passband. $S_{23}$ in the frequency range of 2.401~2.483 GHz is less than −20 dB. However,$S_{11}$ is −5 dB in the frequency range of 2.401~2.483 GHz because there is an impedance mismatch due to packaging effects). 

For the UHB + MHB diplexer, the product specifications are shown in Figure 17 and Table 8, and the $S_{31}$ performance for the passband (WIFI 6E band) is excellent. For $S_{21}$, the measured and simulation results in the passband (2.4 GHz WIFI band) differ by 1.2 dB. ($S_{21}$ is −3.2 dB in the passband and −20 dB in the stopband. $S_{31}$ is −2 dB in the passband and −40 dB in the stopband.) The return loss and isolation, which are shown in Figure 17c,d, are respectively greater than 10 dB and 25 dB in the passbands for the 2.4 GHz WIFI and WIFI 6E bands.

## 5. Conclusions

This study designs antenna-plexers using SAW resonators and considers the design of the inductor in the matching circuits. The results show that designs based on improved matching inductors give better results for various application scenarios. Methods and guidelines are proposed to synthesize a SAW extractor and a diplexer that combine different frequency bands. The final design meets the specifications by using an IDT structure and packaging. The measured results agree with those for the simulated data. However, there are still limitations and areas for improvement. Further study is required to develop a better matching inductor design.

## Figures and Tables

**Figure 1 micromachines-15-00089-f001:**
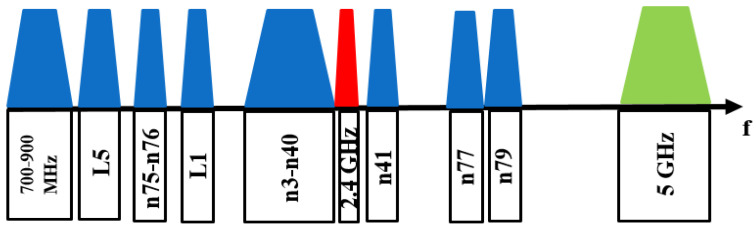
Various frequency bands that are used for handheld devices [1]. The blue colors denote the bands used for cellular communications.

**Figure 2 micromachines-15-00089-f002:**
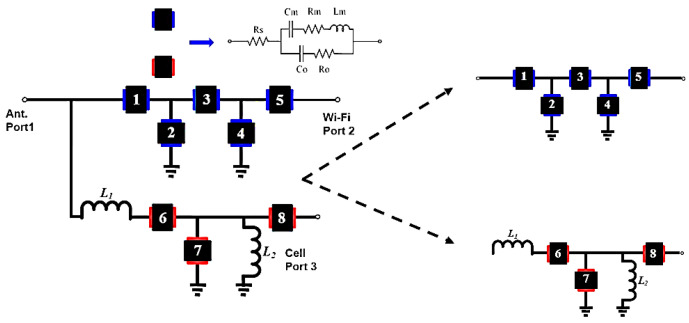
SAW extractor topology. Black boxes with numerical insets represent SAW resonators. Resonators 1 to 5 are used for bandpass filter at WiFi and resonators 6 to 8 for band-stop filter.

**Figure 3 micromachines-15-00089-f003:**
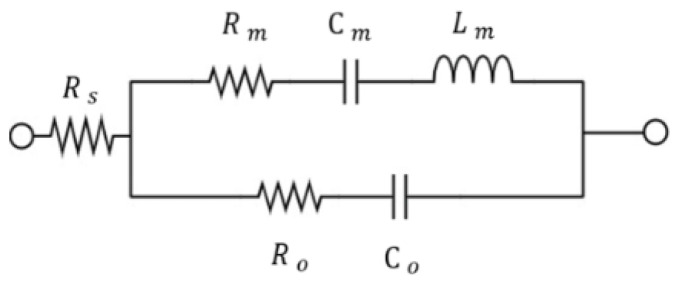
Equivalent mBVD model for a SAW resonator [18].

**Figure 4 micromachines-15-00089-f004:**
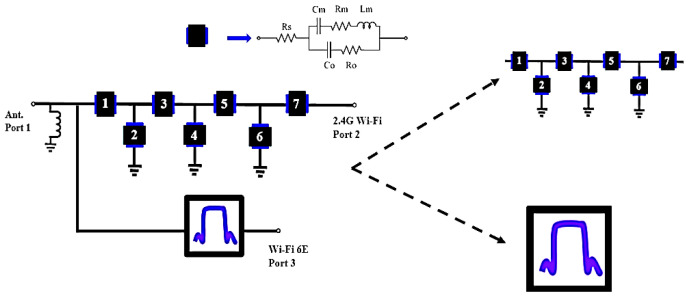
UHB + MHB diplexer topology. Black boxes with numerical insets represent SAW resonators. Resonators 1 to 7 are used for bandpass filter at mid-high band.

**Figure 5 micromachines-15-00089-f005:**
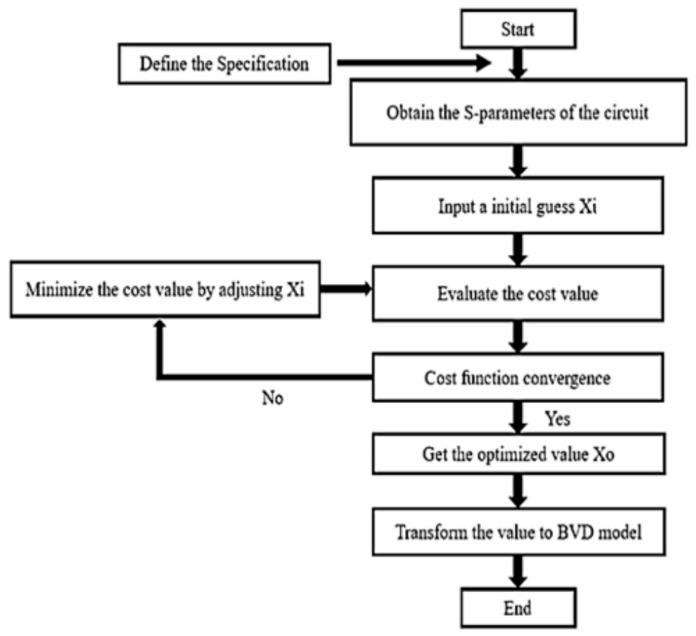
Flowchart for the antenna-plexer design.

**Figure 6 micromachines-15-00089-f006:**
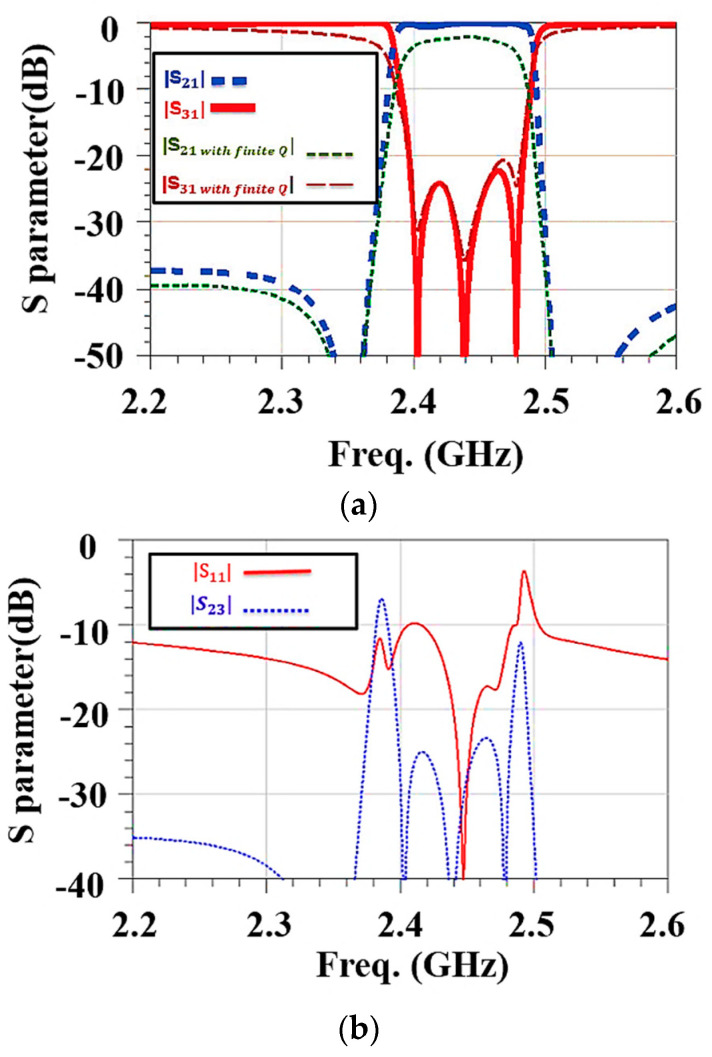
Frequency responses for a SAW extractor using the mBVD model. (**a**) Insertion loss and (**b**) return loss and isolation.

**Figure 7 micromachines-15-00089-f007:**
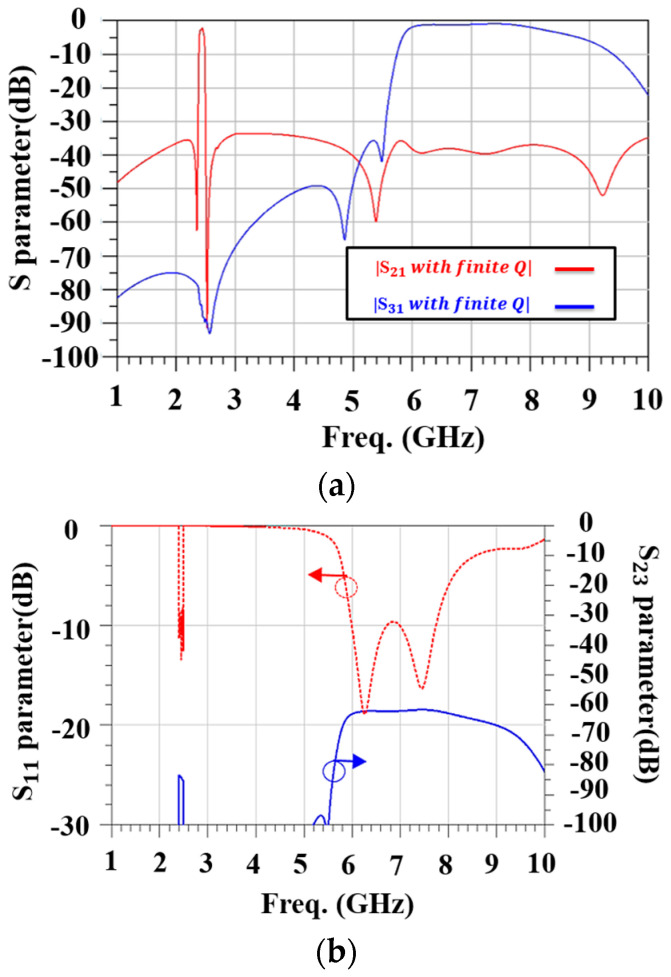
Frequency responses for a UHB + MHB diplexer using the mBVD model. (**a**) Insertion loss and (**b**) isolation.

**Figure 8 micromachines-15-00089-f008:**
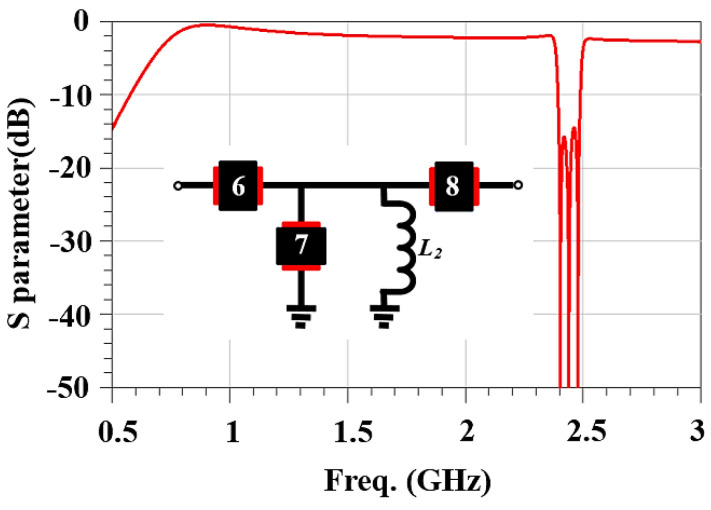
The effects of a parallel inductor on the frequency response for the band-stop filter. Black boxes represent SAW resonators and the numerical insets correspond to those in Figure 2.

**Figure 9 micromachines-15-00089-f009:**
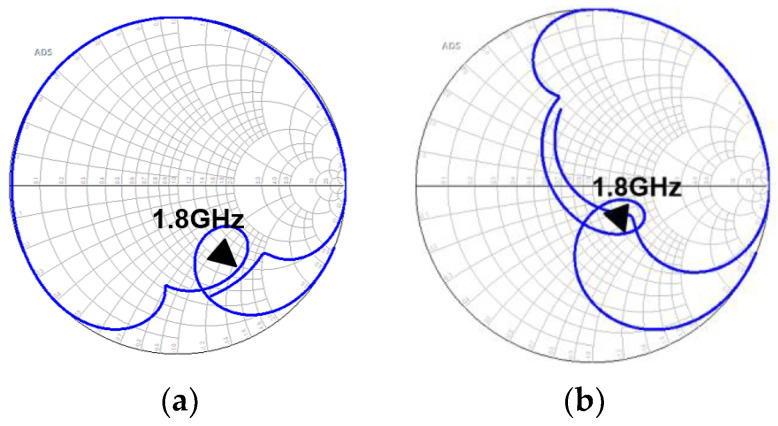
The Smith chart for the band-stop filter $S_{11}$ (**a**) without a series inductor and (**b**) with a series inductor.

**Figure 10 micromachines-15-00089-f010:**
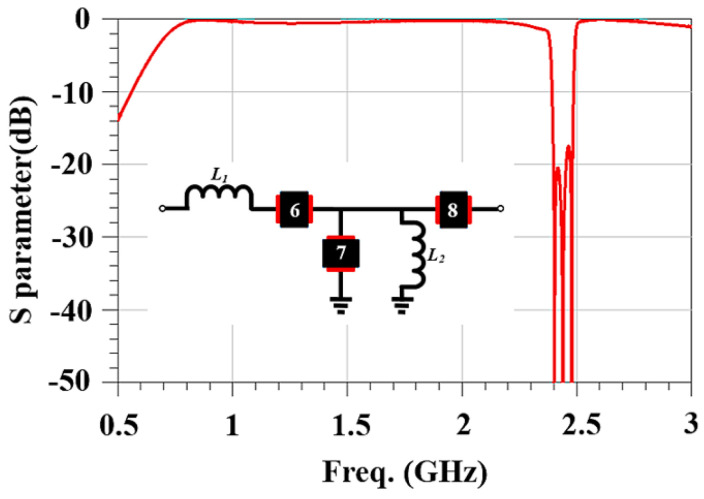
The frequency response for the band-stop filter if a series inductor is added for matching. Black boxes represent SAW resonators and the numerical insets correspond to those in Figure 2.

**Figure 11 micromachines-15-00089-f011:**
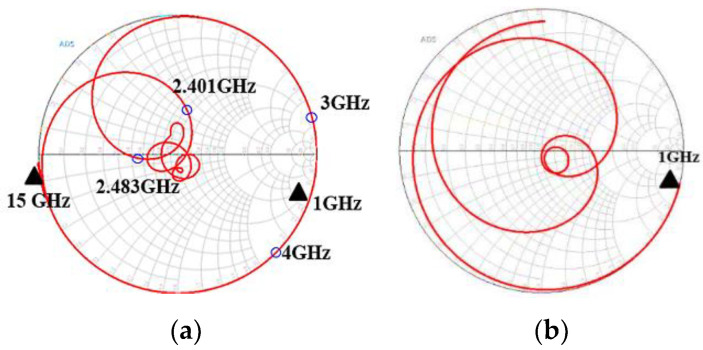
The Smith chart for $S_{11}$ for (**a**) a 7th-order SAW filter and (**b**) a WIFI 6E filter.

**Figure 12 micromachines-15-00089-f012:**
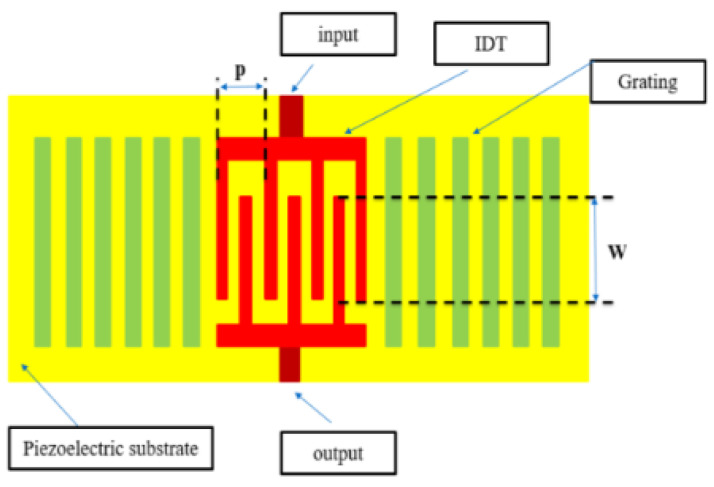
Structure of a single-port resonator.

**Figure 13 micromachines-15-00089-f013:**
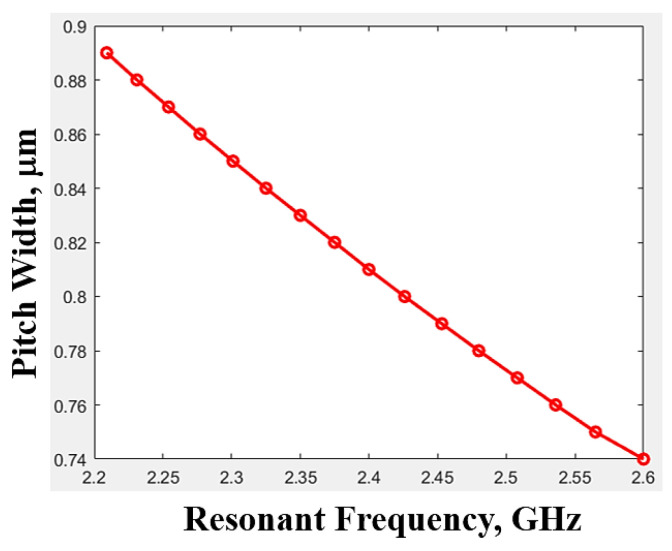
The relationship between pitch width (*p*) and $f_{s}$.

**Figure 14 micromachines-15-00089-f014:**
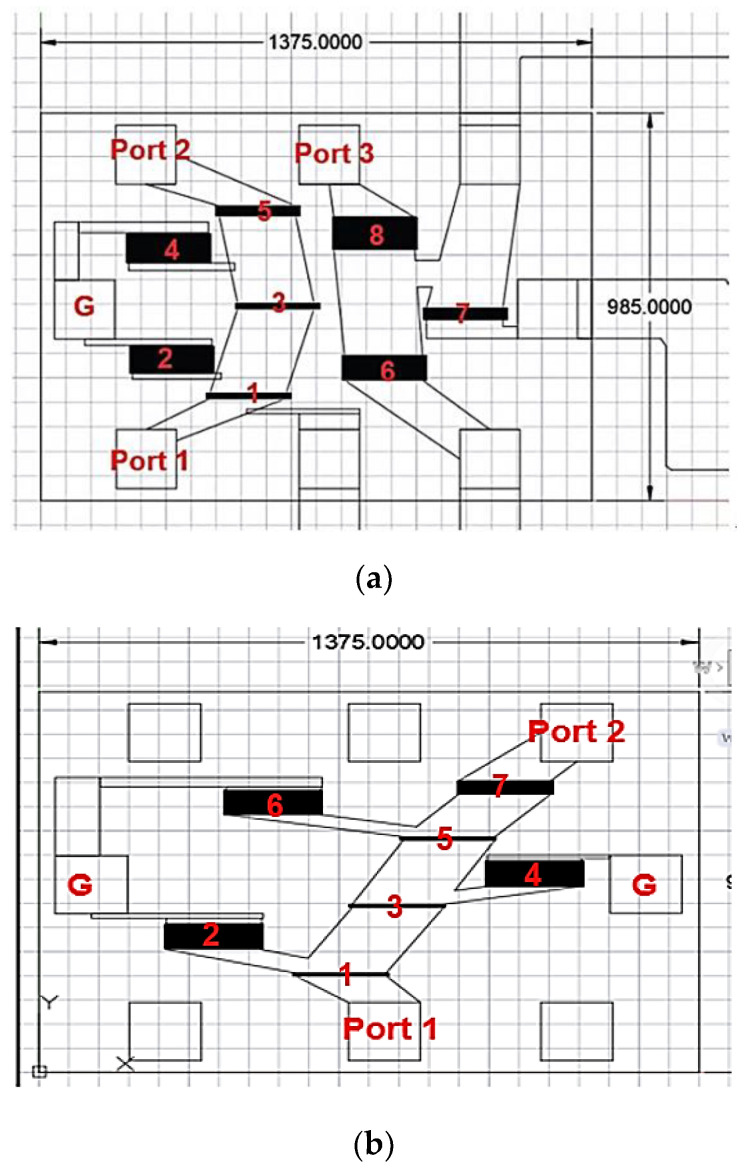
Circuit layout of (**a**) SAW extractor and (**b**) UHB + MHB diplexer. The unit is µm. The black boxes represent SAW resonators and the numerical insets correspond to those in Figure 2 and Figure 4, respectively.

**Figure 15 micromachines-15-00089-f015:**
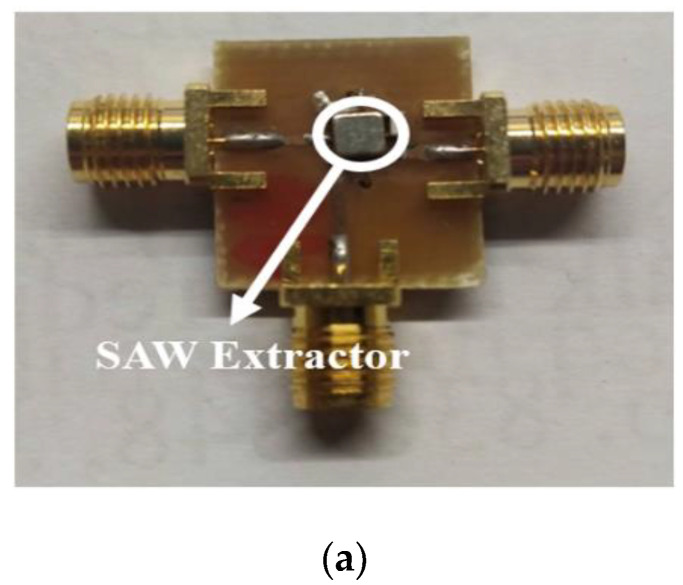
Realized circuit for (**a**) SAW extractor and (**b**) UHB + MHB diplexer.

**Figure 16 micromachines-15-00089-f016:**
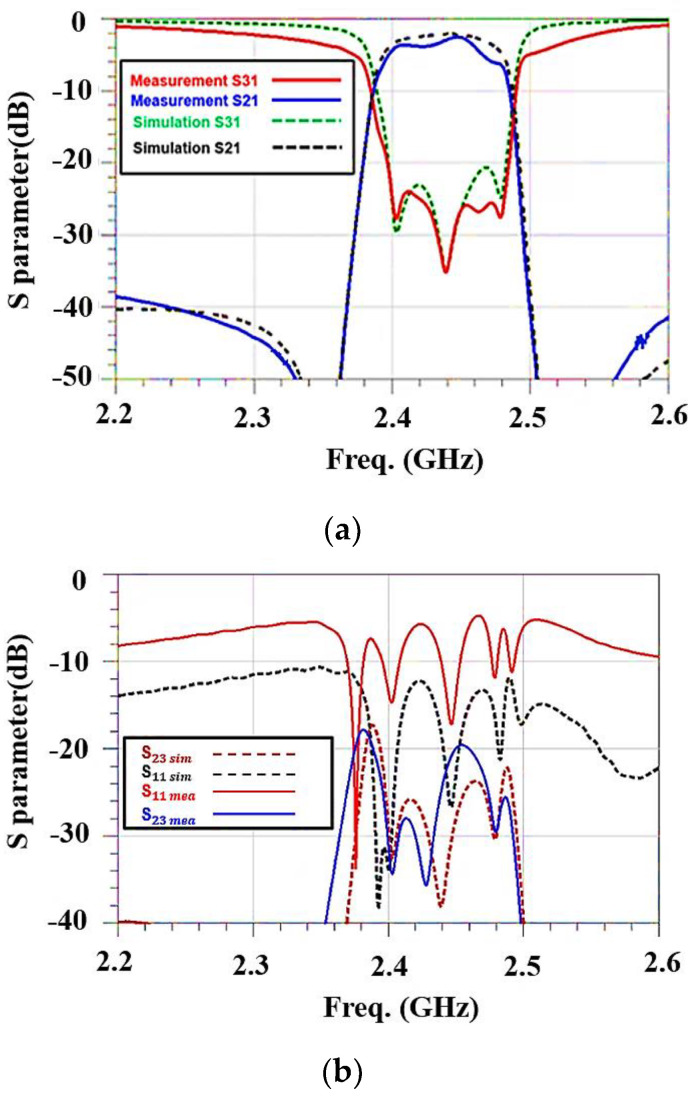
Comparison of experimental and simulated results for SAW extractor: (**a**) $S_{21}$ and $S_{31}$ and (**b**) $S_{11}$ and $S_{23}$.

**Figure 17 micromachines-15-00089-f017:**
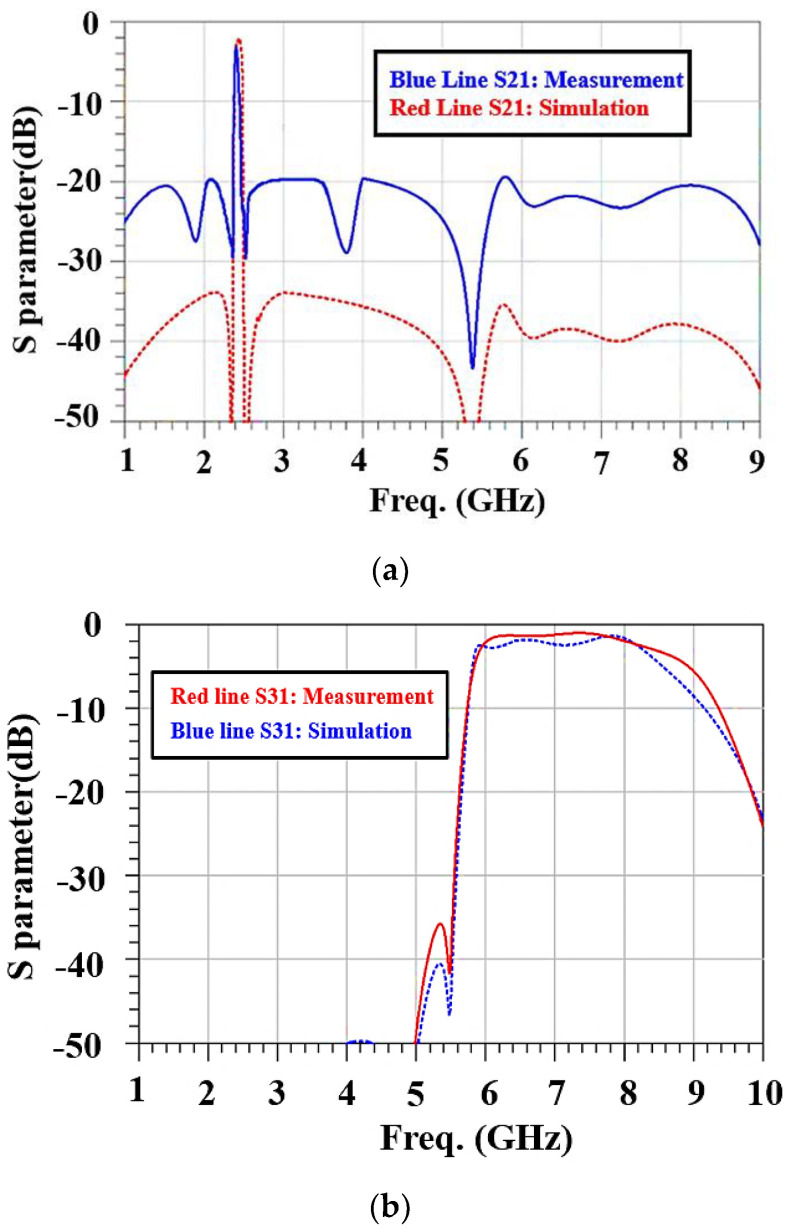
Comparison of experimental and simulated results for UHB + MHB diplexer: (**a**) $S_{21}$, (**b**) $S_{31}$, (**c**) $S_{11}$, and (**d**) $S_{23}$.

**Table 1 micromachines-15-00089-t001:** ${Ŝ}_{31}\;$ and ${Ŝ}_{11}$ in frequency ranges $f_{in}$ and $f_{out}$.

	$f_{in}$	$f_{out}$
${Ŝ}_{21}$	−1.5 dB	−38 dB
${Ŝ}_{31}$	−20 dB	−1.5 dB
${Ŝ}_{11}$	−10 dB	−10 dB

**Table 2 micromachines-15-00089-t002:** Targets for ${Ŝ}_{21}$,${Ŝ}_{11}\;$ in frequency ranges $f_{in1}$ and $f_{out1}$.

	$f_{in1}$	$f_{out1}$
${Ŝ}_{21}$	−1 dB	−35 dB
${Ŝ}_{11}$	−10 dB	−1 dB

**Table 3 micromachines-15-00089-t003:** Optimized BVD parameters for SAW extractor.

BVD Element	$L_{m}$ (nH)	$C_{m}$ (pF)	$C_{0}$ (pF)	${k_{t}}^{2}$ (%)
1	94.8143	0.04458	0.6863	6.1
2	22.7114	0.20072	3.0898	6.1
3	120.7604	0.03523	0.5423	6.1
4	31.7442	0.14447	2.2239	6.1
5	43.2109	0.09893	1.5229	6.1
6	37.1094	0.11836	1.8219	6.1
7	51.0005	0.08349	1.2853	6.1
8	23.1932	0.20144	3.1008	6.1
*L* _1_	4.7	*L* _2_	6.8	

**Table 4 micromachines-15-00089-t004:** Optimized BVD parameters for UHB + MHB diplexer.

BVD Element	$L_{m}$ (nH)	$C_{m}$ (pF)	$C_{0}$ (pF)	${k_{t}}^{2}$ (%)
1	187.3719	0.02235	0.3382	6.1
2	22.5695	0.2006	3.0733	6.1
3	173.0049	0.02451	0.3683	6.1
4	22.1642	0.2038	3.1276	6.1
5	163.6990	0.02591	0.3893	6.1
6	23.5616	0.1930	2.9921	6.1
7	38.6325	0.1111	1.6609	6.1
*L*	3.3			

**Table 5 micromachines-15-00089-t005:** Structure parameters for SAW extractor.

BVD Element	*p* (μm)	*W* (μm)	$N_{t}$	$N_{p}$
1	0.8388	99	200	20
2	0.7951	28	200	20
3	0.8092	82	200	20
4	0.7918	8.4	200	20
5	0.8272	50.8	200	20
6	0.7948	10	200	20
7	0.8300	52.3	200	20
8	0.7948	24.6	200	20

**Table 6 micromachines-15-00089-t006:** Structure parameters for diplexer.

BVD Element	*p* (μm)	*W* (μm)	$N_{t}$	$N_{p}$
1	0.7878	6.8	200	20
2	0.8240	64	200	20
3	0.7934	75	200	20
4	0.8230	65	200	20
5	0.7934	8	200	20
6	0.8260	62	200	20
7	0.7988	34	200	20

**Table 7 micromachines-15-00089-t007:** Comparison of the results by IDT simulation, experimental data, and MURATA product [19].

		*f* = 2.442 GHz	*f* = 2.37 GHz	*f* = 2.555 GHz
$S_{21}$ (dB)	IDT	−1.7	−42	−52
	Meas.	−3.2	−44	−54
	Murata	−1	−45	−55
$S_{31}$ (dB)	IDT	−25	−2	−0.8
	Meas.	−24	−4	−1.7
	Murata	−21	−1.5	−0.67
$S_{11}$ (dB)	IDT	−12	−12	−18
	Meas.	−8	−8	−8
	Murata	−10.9	−15.5	−15.5
$S_{23}$ (dB)	IDT	−32	−34	−40
	Meas.	−19	−22	−42
	Murata	−22	−47	−47

**Table 8 micromachines-15-00089-t008:** Comparison of the results by IDT simulation, experimental data, and Qorvo product [20].

		$\mathrm{IL}@\;=(f_{c}$ *)*	$\mathrm{RL}@\;=(f_{c}$ *)*
$S_{21}$(dB)	IDT	−2	−8
	Meas.	−3.2	−9
	Qorvo	−1	−12
$S_{31}$(dB)	Meas.	−2	−12
	Qorvo	−1	−13

## Data Availability

The data presented in this study are available on request from the corresponding author.

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
