# Peer review of "Design for SAW Antenna-Plexers with Improved Matching Inductance Circuits"

_micromachines, 2023, doi:10.3390/mi15010089_

Round 1

Reviewer 1 Report

Comments and Suggestions for Authors

The study presents a comprehensive design approach for antenna-plexers, incorporating a surface acoustic wave (SAW) Extractor and a UHB+MHB diplexer for LTE 4G and 5G bands. The integration of multiple functionalities in a compact unit is commendable. However, there are areas where certain improvements can enhance the overall quality of the article.

comments:

  1. The article mentions that the SAW Extractor utilizes a transformation to convert the band-stop filter into a high-pass filter at low frequencies and a capacitor at high frequencies. Could you provide more details about the specific transformation technique employed and how it affects the performance of the Extractor?
  2. The literature review is weak and does not cover the relevant state-of-the-art. This aspect needs improvement.In the realm of antenna design, there has been a surge of recent studies and intriguing concepts. Incorporating these articles into the introduction will greatly enhance the depth of the article.

[1] 10.1109/ACCESS.2023.3238571

[2] https://doi.org/10.3390/app13074380

3.        The design goals for the SAW Extractor and the UHB+MHB diplexer are briefly mentioned, but the specific criteria and constraints used for optimization are not elaborated. Could you provide more information on the specific design objectives and cost functions employed during the optimization process?

4.        The experimental results are mentioned to be in good agreement with the simulation results for the SAW Extractor. Could you provide more details about the experimental setup and measurement techniques used to validate the performance of the Extractor? Additionally, were any specific challenges encountered during the experimental characterization?

5.        I am in complete agreement with the concept of the LC model proposed in the article. However, it would be beneficial to provide a suggested numerical range for the element values of this model. If this information is not available, a more comprehensive description would be acceptable. Additionally, incorporating the modeling techniques employed in the referenced articles would undoubtedly enhance the completeness of your model.

[3] 10.3906/elk-1609-130

[4] S. Majidifar, “Design of High Performance Miniaturized Lowpass Filter Using New Approach of Modeling”, ACES Journal, vol. 31, no. 01, pp. 52–57, Aug. 2016.

  1. The theoretical analysis of the frequency response is briefly mentioned, but the specific methodology used to solve the impedance matrix for the SAW resonators is not described (enough). Could you provide more information about the theoretical analysis technique employed and how it contributes to ensuring the desired frequency response characteristics?
  2. The quality of the pics are sub-standard.
Comments on the Quality of English Language

none

Author Response

Description of Revisions in Response to the Reviewers Comments

We are indebted to the reviewers for the valuable comments. Accordingly, we have made revisions in the revised manuscript with revisions marked in yellow. The specific revisions are described as belows.

Reviewer :

The study presents a comprehensive design approach for antenna-plexers, incorporating a surface acoustic wave (SAW) Extractor and a UHB+MHB diplexer for LTE 4G and 5G bands. The integration of multiple functionalities in a compact unit is commendable. However, there are areas where certain improvements can enhance the overall quality of the article.

comments:

  • The article mentions that the SAW Extractor utilizes a transformation to convert the band-stop filter into a high-pass filter at low frequencies and a capacitor at high frequencies. Could you provide more details about the specific transformation technique employed and how it affects the performance of the Extractor?

<Reply>

The band-stop filter functions as a stop band at the design frequency near 2.442 GHz. Outside of the stop band, consider the frequency behavior of the filter in the low-frequency range of 0.7 to 2.3 GHz and the high-frequency range of 2.484 to 2.7 GHz. In both frequency ranges, resonators 6, 7, and 8 behave as capacitive. In the low-frequency range, the resonator 7 in shunt with inductor L2 is inductive and the setup behaves like a high-pass filter. In the high-frequency range, it behaves like a capacitor and the setup behaves like a capacitor.

<Action>

Please refer to lines 4-6 in the abstract in the revised manuscript. A more detailed description can be found in lines 1-6 of the 3rd paragraph in Section III.C.

  • The literature review is weak and does not cover the relevant state-of-the-art. This aspect needs improvement.In the realm of antenna design, there has been a surge of recent studies and intriguing concepts. Incorporating these articles into the introduction will greatly enhance the depth of the article.

[1] 10.1109/ACCESS.2023.3238571

[2] https://doi.org/10.3390/app13074380

<Reply>

Thanks for the comment. However, the two papers recommended by the reviewer are not related to antenna-plexers, and are therefore not included in the manuscript. Current research focuses on Surface Acoustic Wave (SAW) filters rather than the antenna design. Nonetheless, we have added 10 relevant papers in the references.

<Action>

Please refer to references [7]-[16] in the revised manuscript.

  • The design goals for the SAW Extractor and the UHB+MHB diplexer are briefly mentioned, but the specific criteria and constraints used for optimization are not elaborated. Could you provide more information on the specific design objectives and cost functions employed during the optimization process?

<Reply>

The parameters are defined. For example, , , and are the number of frequency points for , , and  at intervals of 1MHz. The optimization process is mentioned clearly. For practical specifications, the cost function is constructed in two parts: the in-band and out-band frequencies. Both entail subtracting the S-parameters over the frequency range of the antenna-plexers from the target specifications. If the result exceeds zero, the specification is not met, necessitating the cost function calculation. Use the pattern search method in the MatLab to iterate the calculation until the result is satisfactorily close to zero [8], meaning it meets the specifications.

<Action>

Please refer to the 2nd paragraph of Section II.D in the revised manuscript.

  • The experimental results are mentioned to be in good agreement with the simulation results for the SAW Extractor. Could you provide more details about the experimental setup and measurement techniques used to validate the performance of the Extractor? Additionally, were any specific challenges encountered during the experimental characterization?

<Reply>

More details on experimental setup and measurement procedure are included. In the experiment, the SOLT calibration method is used to calibrate the Vector Net-work Analyzer (VNA). The VNA is then connected with a cable to the packaged experimental prototype. Therefore, the measurement results include the degradations and other parasitic effects from the interconnects in the package and transmission lines in the test board

<Action>

Please refer to lines 1-5 in the 1st paragraph of Section IV.C in the revised manuscript.

  • I am in complete agreement with the concept of the LC model proposed in the article. However, it would be beneficial to provide a suggested numerical range for the element values of this model. If this information is not available, a more comprehensive description would be acceptable. Additionally, incorporating the modeling techniques employed in the referenced articles would undoubtedly enhance the completeness of your model.

[3] 10.3906/elk-1609-130

[4] S. Majidifar, “Design of High Performance Miniaturized Lowpass Filter Using New Approach of Modeling”, ACES Journal, vol. 31, no. 01, pp. 52–57, Aug. 2016.

<Reply>

Designing  and the resonant frequency requires adherence to two rules: the resonant frequency fs = 1/(2π*√Lm/√Cm) and Zm = √Lm/√Cm  resides within the specified range of 50 to 10000 ohms. Given these two values, the inductance (Lm) and capacitance (Cm) can be obtained. In numerical optimization, Zm= 50 ohms is usually set as the initial value for iteration.

<Action>

Please refer to the last 5 lines in the 2nd paragraph of Section II.E. The paper [4] suggested here is cited in the 3rd paragraph of Section 1 and referred as [16] in the revised manuscript.

  • The theoretical analysis of the frequency response is briefly mentioned, but the specific methodology used to solve the impedance matrix for the SAW resonators is not described (enough). Could you provide more information about the theoretical analysis technique employed and how it contributes to ensuring the desired frequency response characteristics?

<Reply>

Each element in the impedance matrix can be observed from (1), and  can be determined by the following expression:

 =

By definition, it can be calculated by applying the input current  to the j-th port while keeping all remaining ports open-circuited, and measuring the open-circuit voltage at the i-th port.

<Action>

Please find the 2nd paragraph of Section II.C in the revised manuscript.

  • The quality of the pics are sub-standard.

<Reply>

Thanks for the comment. Most figures have been re-drawn with better quality.

<Action>

Please find the upgraded figures in Figs. 1, 6(a)-(b), 7(a)-(b), 8, 10, 13, 17 (a)-(d), and Tables VII, VIII in the revised manuscript.

Reviewer 2 Report

Comments and Suggestions for Authors

This study designs antenna-plexers, including a surface acoustic wave (SAW) Extractor and a UHB+MHB duplexer, for LTE 4G and 5G bands. Experimental results obtained are in good agreement with the simulation results.  The study uses an optimization approach, developed by Authors earlier in [7], to design the antenna-plexer module, which is composed of surface acoustic wave (SAW) resonators and inductors for small-sized, high-efficiency RF modules in handsets.

The manuscript is relevant for the field but not presented in acceptable manner. The number of cited references is very small and does not reflect an importance of subject. The manuscript’s results look reproducible based on the details given in the methods section. The figures/tables are of very low quality and do not properly show the data. The data are interpreted appropriately and consistently throughout the manuscript.

I have several comments:

a) The introduction poorly reflects the literature data on methods for calculating various devices with SAW and what place among them the optimization approach developed by the Authors occupies. Most number of references are the proceedings of various meetings, rather than ordinary articles and books.

b) The figures are of very low quality and are located significantly above the parts of the article where they are explained. Caption for fig. 13 does not explain the content and meaning of some curves. In some figures, the dimensions along the axes are not indicated.

c) Tables VII–VII are of very low quality. In these tables, the words IDT Simulation and MURATA Product first appear at the end of the article in the tables, but not in the text. The use and purpose of comparing these approaches must be explained.

Additional comments are in attached pdf file.

In conclusion, the Article involves new results but quality of representation is very low and the manuscript needs major revision

Comments on the Quality of English Language

Minor editing of English language required

Author Response

Description of Revisions in Response to the Reviewers Comments

We are indebted to the reviewers for the valuable comments. Accordingly, we have made revisions in the revised manuscript with revisions marked in yellow. The specific revisions are described as belows.

  1. The introduction poorly reflects the literature data on methods for calculating various devices with SAW and what place among them the optimization approach developed by the Authors occupies. Most number of references are the proceedings of various meetings, rather than ordinary articles and books.

<Reply>

There is a lot of literature on the miniaturization of antenna-plexers, mainly on microstrip or LTCC. We cite quite a few papers in this regard, including one suggested by Reviewer 1. However, the antenna-plexers using SAW are more involved in the industry, not so many open literature. To emphasize the importance of this study, we quote a few sentences from [2]. “Various applications use different frequency ranges, leading to diverse system requirements for each band. Crafting an effective filter module involves ensuring that its frequency response meets specified requirements across all ranges. Therefore, intelligent design considerations, including topology selection, optimizing SAW resonator characteristic impedance, and careful material choice, are crucial [2].”

<Action>

Please find the new references [7]-[16] and lines 3-7 in the 4th paragraph of the Introduction Section in the revised manuscript. 

  1. The figures are of very low quality and are located significantly above the parts of the article where they are explained. Caption for fig. 13 does not explain the content and meaning of some curves. In some figures, the dimensions along the axes are not indicated.

<Reply>

Most figures have been re-drawn with better quality. They are located close to the article where they are explained. We have inserted new text to explain Fig. 13. The relationship between pitch (p) and  is plotted using experimental data which was measured by the manufacturer. The piezoelectric material for this study is , and the electrode thickness is 0.17 µm, IDT pair is 200, grating order is 20, aperture length W is 50 µm, and the resonator experimental data for pitch width p ranging from 0.89 µm to 0.74 µm (in 0.01 µm intervals).

<Action> Please find the first paragraph of Section IV.A in the revised manuscript. Also find the upgraded figures in Figs. 1, 6(a)-(b), 7(a)-(b), 8, 10, 13, and 17 (a)-(d).

  1. Tables VII–VIII are of very low quality. In these tables, the words IDT Simulation and MURATA Product first appear at the end of the article in the tables, but not in the text. The use and purpose of comparing these approaches must be explained.

<Reply>

Tables VII and VIII have been modified, providing the performance of IDT and Murata, respectively, in terms of simulated Q values and product specifications. Explanatory notes have been added to the tables.

<Action>

Please refer to Section IV.C in the revised manuscript.

Round 2

Reviewer 1 Report

Comments and Suggestions for Authors

None

Comments on the Quality of English Language

None

Reviewer 2 Report

Comments and Suggestions for Authors

The Authors answered all questions and comments. Minor corrections that need to be made are shown in the comments in the pdf file.

The Article can be published after minor revision.

Comments on the Quality of English Language

Minor editing of English language required